# Cancer Is Associated with the Emergence of Placenta-Reactive Autoantibodies

**DOI:** 10.3390/biomedicines11020316

**Published:** 2023-01-23

**Authors:** Sara Khorami Sarvestani, Sorour Shojaeian, Ramin Sarrami-Forooshani, Mir Saeed Yekaninejad, Kambiz Gilany, Abbas Ghaderi, Maryam Hashemnejad, Asiie Olfatbakhsh, Farzane Notash Haghighat, Samaneh Montazeri, Allan Stensballe, Mahmood Jeddi-Tehrani, Amir-Hassan Zarnani

**Affiliations:** 1Reproductive Immunology Research Center, Avicenna Research Institute, ACECR, Tehran 1936773493, Iran; 2Monoclonal Antibody Research Center, Avicenna Research Institute, ACECR, Tehran 1936773493, Iran; 3Department of Biochemistry, School of Medical Sciences, Alborz University of Medical Sciences, Karaj 3149969415, Iran; 4Department of ATMP, Breast Cancer Research Center, Motamed Cancer Institute, ACECR, Tehran 1517964311, Iran; 5Department of Epidemiology and Biostatistics, School of Public Health, Tehran University of Medical Sciences, Tehran 1417613151, Iran; 6Shiraz Institute for Cancer Research, School of Medicine, Shiraz University of Medical Sciences, Shiraz 71345319, Iran; 7Obstetrics and Gynecology Department, School of Medical Sciences, Alborz University of Medical Sciences, Karaj 3134877179, Iran; 8Breast Cancer Research Center, Motamed Cancer Institute, ACECR, Tehran 1517964311, Iran; 9Department of Health Science and Technology, Aalborg University, 9220 Aalborg, Denmark; 10Department of Immunology, School of Public Health, Tehran University of Medical Sciences, Tehran 1417613151, Iran

**Keywords:** autoantibody, antigen, cancer, placental proteins, pregnancy

## Abstract

Placenta-specific antigens are minimally expressed or unexpressed in normal adult tissues, while they are widely expressed in cancer. In the course of carcinogenesis, a vast array of autoantibodies (AAbs) is produced. Here, we used a quantitative approach to determine the reactivity of AAbs in the sera of patients with breast (BrC: N = 100, 100% female, median age: 51 years), gastric (GC: N = 30, 46.6% female, median age: 57 years), bladder (BC: N = 29, 34.4% female, median age: 57 years), and colorectal (CRC: N = 34, 41.1% female, median age: 51 years) cancers against first-trimester (FTP) and full-term placental proteome (TP) in comparison with age- and sex-matched non-cancer individuals. Human-on-human immunohistochemistry was used to determine reactive target cells in FTP. The effect of pregnancy on the emergence of placenta-reactive autoantibodies was tested using sera from pregnant women at different trimesters of pregnancy. Except for BC, patients with BrC (*p* < 0.0284), GC (*p* < 0.0002), and CRC (*p* < 0.0007) had significantly higher levels of placenta-reactive AAbs. BrC (*p* < 0.0001) and BC (*p* < 0.0409) in the early stages triggered higher autoantibody reactivity against FTP. The reactivities of BrC sera with FTP did not show an association with ER, PR, or HER2 expression. Pregnancy in the third trimester was associated with the induction of TP- and not FTP-reactive autoantibodies (=0.018). The reactivity of BrC sera with placental proteins was found to be independent of gravidity or abortion. BrC sera showed a very strong and specific pattern of reactivity with scattered cells beneath the syncytiotrophoblast layer. Our results reinforce the concept of the coevolution of placentation and cancer and shed light on the future clinical application of the placental proteome for the non-invasive early detection and treatment of cancer.

## 1. Introduction

For more than a century, analogies between cancer and trophoblastic cell behavior during placentation have been mentioned [1]. Cancer cells and placenta share some characteristics, including immune escape and growth mechanisms as well as antigenic determinants [2,3]. These include proteins that regulate and complement apoptosis regulatory mechanisms, growth factors that are involved in angiogenesis, and large numbers of tumor-associated antigens [4,5]. Shared antigens between cancer cells and the placenta have led to the idea that cancer therapy may be possible through vaccination with placental antigens that may act as a multi-epitope vaccine. This idea is well supported by immune-placental therapy (IPT) for cancer [6,7]. It is suggested that immunization with placental antigens during pregnancy accounts for the reduction in cancer risk in women with prior pregnancies. This information is in line with and supported by further studies that led to the concept of “cancer stem cell therapy” [8,9,10,11].

Due to the hemochorial nature of human placentation, placental antigens are continuously presented to the maternal immune system and act as a source of fetal antigen exposure to the mother [11]. Placenta-specific antigens are minimally expressed or unexpressed in normal adult tissue, while they are widely expressed during cancer [12].

Any alteration in metabolism and/or physiological conditions is reflected by a shift in the proteome pool [13,14], which could result in a change in the autoantibody (AAb) profile. Each individual has a pool of autoantibodies, the composition of which is dramatically changed depending on the physiological condition. These autoantibodies are produced as a result of altered protein expression, the emergence of neoantigens, inflammation, and defective immune tolerance. Among the major functions exerted by AAbs are the discrimination of diseased cells from normal ones and the maintenance of host homeostasis [15]. Cancer is a pathological condition in which the composition of the autoantibody repertoire is changed and, in this regard, monitoring serum AAbs directed against tumor-associated antigens is considered a promising approach for early cancer diagnosis [16]. Deviations in the serum contents of AAbs in different diseases, especially cancers, can be detected much earlier than when the clinical manifestations appear and can be used for the early detection of the disease [17]. The screening of cancer by cancer-associated AAbs is preferred to that of the corresponding antigens. This is because of the biological amplification, early response, and stability of the antibodies [18]. However, harnessing autoantibodies for cancer screening has faced several drawbacks. In general, 10–30% of cancer patients develop a specific antibody response against a defined tumor-associated antigen, which could be due to the heterogeneous nature of cancer, where the repertoires of antigens vary extensively among tumors [19]. This is evidenced by the aberrant processing or regulation of different proteins in patients suffering from the same cancer [20]. Consequently, patients express heterogeneous repertoires of AAbs. In line with this notion, the frequency of antibodies against a defined cancer-associated antigen ranges between 1 to ∼15% [21]. Interestingly, cancer patients mainly develop autoantibodies against antigens that are abundantly expressed in the placenta [22]. Antibodies against survivin, PLAC1, livin, and NY-ESO-1 are among the cancer-associated autoantibodies for which the corresponding antigens are highly expressed in the placenta [23,24,25,26,27].

There are many features that placentation and cancer have in common. This study was conducted to examine antigenic analogies between placenta and cancers of the breast, stomach, bladder, and colon by testing the presence of placenta-reactive autoantibodies in the sera of cancer patients. Indeed, examining the potential effect of pregnancy on the emergence of placenta-reactive AAbs was also among the additional aims of this study.

## 2. Materials and Methods

### 2.1. Placental Samples

In this study, we collected four normal human first-trimester whole placentas and four full-term placentas with the inclusion and exclusion criteria reported earlier [28]. All procedures were carried out following approval by the ethical committee of Avicenna Research Institute (ARI) (ethical approval No: 1397.007) and in accordance with the revised version of the Helsinki Declaration in 2013. The healthy pregnant women who voluntarily terminated an unplanned pregnancy provided normal FT placentas. At the cesarean section, TPs were collected. Throughout the course of the pregnancy, all aspects of pregnancy health, including systolic blood pressure, body mass index (BMI), mother and fetus weights, blood glucose levels, and obesity, were regularly tracked. None of the pregnant women had ever aborted a child, suffered from a chronic or serious disease, or abused medicine prior to the cesarean section or induced abortion. All TPs belonged to male fetuses to limit the impact of sex on the proteome profile; however, the genders of terminated fetuses (gestational age 12 weeks) were unknown. When compared to TP placentas, FT placentas had mean gestational ages of 10 ± 2 weeks and 38 ± 1 week, respectively. A pathologist examined each placenta sample and determined that they were all normal. The placentas were swiftly brought to the laboratory in cold phosphate-buffered saline (PBS). A sterile scalpel was used to cut and pool five samples from each placenta: four samples of 1 cm diameter (each about one gram) from four locations (including the maternal and fetal sides), and one sample from the center. The combined samples were then washed three times in cold PBS to remove contaminating blood, aliquoted, and stored in liquid nitrogen until protein extraction.

### 2.2. Protein Extraction and Quantification

In order to create the FT and TP pools, four frozen first-trimester placenta samples and four frozen term placenta samples were individually blended. By employing a cooled mortar and pestle and cryogenic grinding with liquid nitrogen, the combined samples were ground into powder. The pulverized sample (0.1 gr) was homogenized in 1 mL of lysis buffer, which contained 8 M urea, 2% CHAPS, and 2% dithiothreitol (DTT) in 5 mM Tris-HCl pH 7.6, and was then incubated for 15 min on ice with gentle vortexing. At 4 °C, the homogenates were centrifuged for 1 h at 15,000× *g*. The supernatants were gathered, and a 2-D Quant Kit was used to measure the protein quantities (GE Healthcare, city name: Chicago, IL, USA).

### 2.3. Serum Samples

Bladder cancer (BC), gastric cancer (GC), colorectal cancer (CRC), non-CRC, and non-BC serum samples were collected from the biobank of Cancer Research Institute, Shiraz University of Medical Sciences (ICR), Shiraz, Iran. Breast cancer (BrC) and non-BrC serum samples were collected from Motamed Breast Cancer Institute (MCI), Tehran, Iran. Non-GC serum samples were obtained from the Gastroenterohepatology Research Center (GEHRC), University of Medical Sciences, Shiraz, Iran. The data of patients and their corresponding controls are summarized in Table 1.

Additional control sera were obtained from non-cancer individuals in three groups:(1)Normal males who were referred to the Farmand medical laboratory for a routine checkup.(2)Normal pregnant women in the first and second trimesters of pregnancy with normal screening tests who were referred to the department of Fetal and Maternal Health in the Hope Generation Foundation (HGF), Tehran, Iran.(3)Normal females in the third trimester of pregnancy who were referred to the cesarean section in Kamali hospital, Karaj, Iran (Table 2).

All pregnant women recruited in this study had a normal pregnancy and pregnancy outcome with no history of pregnancy complications such as gestational diabetes, preeclampsia, or thyroid dysfunction (Table 3).

### 2.4. ELISA

Auto-antibodies against placental proteins were quantified using ELISA in flat-bottom polystyrene 96-well Immunoclon-4 HBX microtiter plates (binding capacity 100–200 ng IgG/cm^2^). Different coating concentrations of placental proteins and serum dilutions were first used to achieve the optimal setup of ELISA with the highest signal-to-noise ratio. A polyclonal antibody was generated with the successive subcutaneous immunization of rabbits with FTP proteins, and a purified antibody was used as the positive control in each run of the ELISA test. To test optimal reactivity with either FTP or TP, the immune reactivity of BrC and non-BrC sera with FTP and TP placental lysates were first measured and compared. Based on the higher reactivity of BrC sera with FTP, all subsequent ELISA tests were performed on placenta lysate from FTP. The plates were coated with human placental lysate proteins from either FTP or TP (5 μg/mL) in a coating buffer (phosphate-buffered saline (PBS), pH 7). The plate was incubated overnight at 4 °C. The plates were then washed three times (each time for 1 min) with PBS-Tween 20 (PBST) (pH 7.4; 0.05% Tween), blocked for 90 min at 37 °C with 2.5% bovine serum albumin (BSA) diluted in PBS-T, and washed three times (each time for 1 min) with PBS-T. Then 100 μL of 1:500 diluted serum samples in PBS-T was added to each well and incubated for one hour at room temperature (RT). After three washes with PBS-T, 100 μL of 1:4000 dilution of horse radish peroxidase (HRP)-conjugated sheep anti-human Ig (Sina Biotech, Tehran, Iran) was added and the plates were incubated for 45 min at RT. Then, the plates were washed three times with PBS-T, and color development was carried out by adding 100 μL TMB (Arya Pishtaz Novin, Tehran, Iran) for 10 min at RT in the dark. The enzyme reaction was stopped with 50 μL per well of 10% *v/v* H_2_SO_4_. The optical density was read within 10 min at 450 nm with a reference OD (620 nm) using a Biochrom Anthos 2020 ELISA plate reader. In positive control wells, homemade rabbit anti-FTP and sheep anti-rabbit Ig-HRP were used as primary and secondary antibodies, respectively. Indeed, at each run, a predefined serum sample with good reactivity was run in duplicate as an internal control. Control wells containing only the coating layer and conjugate or pooled serum (non-cancer controls and patients separately) and conjugate (without coating layer) were also included.

To normalize inter-assay variations and to exclude the potential interference of human immunoglobulin contamination of placenta lysate in the ELISA results (interaction of the conjugate with placental immunoglobulin), three types of corrections were performed as follows:L2–L3 correction: the average OD of control wells containing only pooled serum (non-cancer controls and cancer patients separately) and conjugate (without coating layer) were subtracted from the OD of each sample.TP/FTP correction: the OD of control wells containing only the TP coating layer and conjugate (without serum layer) was divided by the OD of wells containing the FTP coating layer and conjugate to obtain the TP correction index. The OD of wells containing TP as a coating layer was then multiplied by this index.Internal control correction: the average OD of internal controls in all ELISA runs was divided by the average OD of internal controls in each ELISA run and the resulting value was multiplied by the OD of each test well.

### 2.5. Immunohistochemistry (IHC)

To validate the results of ELISA and to visualize the localization of the placental antigens reacting with serum autoantibodies, IHC staining was performed. Tissue fragments from the central zone of four placentas were paraffin-embedded and tissue sections (4 µm thick) were prepared. Tissue slides were deparaffinized in xylene (Sigma Aldrich, Oakville, Canada) and rehydrated in graded ethanol. Next, tissue sections were exposed to Tris-EDTA buffer (10 mM Tris base, 1 mM EDTA, 0.05% Tween 20, pH 9) in a 95 °C water bath for heat-induced antigen retrieval. Endogenous peroxidase activity was blocked by immersing the tissue sections in 0.3% H_2_O_2_ for 10 min. After blocking, slides were incubated with pooled sera (BrC early stage: *n* = 70, BrC late stage: *n* = 30, normal male: *n* = 100, and non-breast cancer female = 100) diluted in the primer reagent of the human-on-human IHC staining kit (abcam, Waltham, USA). All immunostaining procedures were performed according to the manufacturer’s instructions. The following variables were first set up before the final experiment: serum dilution, serum incubation time and temperature, and secondary antibody incubation time. Finally, incubation with a 1:20 dilution of sera at 4 °C overnight and 40 min incubation with a human HRP polymer detection system yielded the best results. After the visualization of signals with diaminobenzidine (DAB) for 5 min, slides were washed, rehydrated, stained with hematoxylin, and mounted. For primary antibody, serum samples from breast cancer patients at different stages (early and late), non-breast cancer women, and normal males were used. In negative reagent control slides, serum was omitted in the staining procedure. Moreover, FTP-immunized rabbit hyperimmune serum was used as the primary antibody in positive control slides. These slides were processed as above and probed with the polyclonal antibody for 90 min followed by washing steps and detection with poly-HRP detection system plus (Mouse/Rabbit) (BioVision, Sina biotech, Tehran, Iran).

### 2.6. Statistical Analysis

Statistical analyses were performed using SPSS software version 26. Graphs were plotted using GraphPad Prism version 9.4.0. Normality was checked with normality plots (histogram and Q–Q plot) by visual inspection. The linear relationship between pairs of continuous variables was measured using the bivariate Pearson correlation coefficient. Statistical differences between the means of unrelated groups were compared using a one-way analysis of variance (ANOVA). To explain the relationship between one or more independent variables (gravidity/abortion) on one serum reactivity, multiple linear regression analysis was used. Statistical significance was defined as *p* < 0.05.

## 3. Results

### 3.1. Cancer was Associated with the Generation of Placenta-Reactive Autoantibodies

In this study, the reactivity of sera from patients with GC, BrC, BC, and CRC with placental proteins was assessed and compared with their corresponding well-confirmed age- and sex-matched non-cancer sera. In the first step, the reactivity of sera from BrC patients was investigated against both FTP and TP and based on the results; the reactivity of other cancer sera was only tested against FTP. When the immune reactivity of sera from 100 BrC patients with TP proteins was compared with that of age- and sex-matched non-BrC women, no statistical difference was observed (*p* = 0.09), although the OD values in patient sera tended to be higher. However, patients with BrC showed significantly higher reactivity with FTP (*p* = 0.0284). This trend was highlighted when the immune reactivity of GC and CRC patients’ sera with FTP was evaluated (*p* = 0.0002 and *p* = 0.0007, respectively) (Figure 1). In BC sera, no statistical difference was observed. Moreover, the reactivity of patients’ sera showed no linear correlation with age and no association with the sex of cases.

### 3.2. BrC at Early Stages Triggered Higher Autoantibody Reactivity against First-Trimester Placental Proteins

To compare placenta-reactive autoantibodies against FTP in early and late stages of cancer, each cancer group was divided into two subgroups based on the cancer stage (early: stage I and II, late: stage III, IV). Interestingly, sera from patients with early stages of BrC showed a robust higher reactivity with FTP when compared with sera from patients with higher stages (*p* < 0.0001). This was also the case in BC patients (*p* = 0.0409). In the case of GC and CRC patients, no such difference was observed (Figure 2).

### 3.3. Reactivity of BrC Sera with FTP Did Not Show an Association with ER, PR, or HER2 Expression

To test the potential contribution of ER, PR, and HER2 expression on the reactivity of patient’s sera with FTP, they were categorized based on the immunohistochemical expression of these markers and their reactivity with FTP was compared. Our results showed no significant association between the expression of these different markers and serum immune reactivity (Figure 3). Due to the fact that, except for two BrC samples, all samples were Ki67 positive and no statistical comparison was performed on the potential effect of Ki67 expression and serum immunoreactivity with placental proteins.

### 3.4. Pregnancy in the Third Trimester was Associated with the Induction of TP-Reactive Autoantibodies

Following the observation of the emergence of placenta-reactive autoantibodies in patients with cancer, the immediate question that arose was whether pregnancy itself is associated with the induction of such antibodies. In this regard, the reactivity of sera from pregnant women at different trimesters of pregnancy with FTP and TP was assessed. As controls, the serum of age-matched males and non-BrC/non-pregnant females were also tested. By comparing the serum reactivity of first-trimester (FTS) (*n* = 100), second trimester (STS) (*n* = 100), third trimester (TTS) (*n* = 110), non-BrC/non-pregnant females (NFS) (*n* = 100), and normal males (NMS) (*n* = 100) with TP or FTP proteome, we found that, without considering a past history of pregnancy, pregnancy itself had no effect on the induction of placenta-reactive autoantibodies (Figure 4A). No significant difference in reactivity with either FTP or TP was observed between nulligravidanon-pregnant women and normal males (Figure 4B). Interestingly, pregnancy in the third trimester was associated with the induction of TP- and not FTP-reactive autoantibodies compared to nulligravida non-pregnant women (*p* = 0.018) (Figure 4C). Of note, a comparison of the serum reactivity of first- and third trimester samples showed that first- and third trimester sera had a higher reactivity with corresponding placental proteins (Figure 4C). Sera from primiparous (Figure 4D) or multiparous (Figure 4E) non-pregnant women had comparable levels of FTP- or TP-reactive autoantibodies compared to primigravid (Figure 4D) or multiparous pregnant women at different trimesters of pregnancy (Figure 4E). Regardless of pregnancy trimester, multiparous pregnant women showed higher reactivity with TP compared with nulligravida non-pregnant women (*p* = 0.0167) (Figure 4F).

### 3.5. The Reactivity of BrC Sera with Placental Proteins Was Not Affected by Gravidity or Abortion 

To rule out the potential effect of gravidity or abortion on the higher reactivity of BrC sera with placental proteins, multivariate regression analysis was performed. Since all samples in the BrC group were females with registered reproductive history, the serum reactivity of 100 normal non-BrC and 100 BrC females with FTP proteome was compared. Although we found a significant contribution of the history of pregnancy in the reactivity of sera from BrC and non-BrC females (*p* = 0.007), the higher reactivity of BrC sera compared to non-BrC females was independent of the previous history of pregnancy (*p* = 0.005). Indeed, the previous history of abortion had no effect on serum reactivity with FTP in both BrC and non-BrC groups (Table 4).

### 3.6. Breast-Cancer-Associated Autoantibodies Recognized Stem-Like Cells in the Placenta

To confirm the results of ELISA, the immunohistochemical staining of FTP with sera from BrC, non-BrC, and normal males was performed. Our results showed a very specific pattern of the reactivity of BrC sera with FTP, which was narrowly confined to some cells of syncytiotrophoblasts and specially syncytial nuclear aggregates (SNA). Indeed, some isolated cells beneath the syncytiotrophoblasts were found to be immunoreactive. The frequency of positive cells was small and both cytoplasmic and nuclear patterns of staining were observed (Figure 5). In the case of non-BrC samples, fewer positive cells were also observed. No specific staining in serum samples of normal males was found. The reactivity of sera from late-stage BrC patients was less than that observed in the sera of early-stage BrC patients. In positive control slides, in which polyclonal rabbit anti-placenta serum was used as the primary antibody, a mixed pattern of cytoplasmic and nuclear reactivity in almost all cells of the placenta, including syncytiotrophoblasts, cytotrophoblasts, and extravillous cytotrophoblasts, was observed. In negative control slides (male sera), no reactivity was noted.

## 4. Discussion

The theory of the fetal origin of cancer has long been proposed and much supporting evidence has been reported [29]. A very fascinating connection between placentation and carcinogenesis has been suggested by Wagner et al. Accordingly, a link between species-specific malignancy rate and placentation type is likely related to the level of adapted cis-regulatory mechanisms in the endometrium, which control endometrial resistance to invisibility in eutherian mammals [30]. The results of our study reinforce the concept of the coevolution of placentation by showing the emergence of first-trimester placenta-reactive autoantibodies in cancer patients, particularly in the early stages of Brc. The breast-cancer-associated autoantibodies recognized rare and dispersed cytotrophoblast cells, which might act as placental stem-like cells. This attractive theory is supported by the low incidence of metastatic cancers in hoofed animals. However, the relationship between invasive placentation in mammals with hemochorial placentation and metastatic cancer is usually defined in terms of antagonistic pleiotropy. According to this theory, cancer metastasis co-evolved with the evolution of invasive placentation, with common mechanisms that peruse the same phenotype and behavior. Although positive pleiotropy theory confronts some challenging issues such as the use of shared mechanisms in other body functions such as wound healing and the presence of metastatic cancer in marsupials, it has some supporting evidence [31]. This very simplistic view of the fetal origin of cancer has been the focus of much research. In this context, plenty of antigens shared by the placenta and cancer cells have been introduced so far [1]. These antigens, which are expressed by syncytiotrophoblasts and/or cytotrophoblasts, can induce strong humoral, cellular, and immune responses [12,32].

The identification of molecular markers with the aim of the noninvasive detection and screening of cancer has been the main interest of many scientists. Autoantibodies are among the cancer biomarkers, which are superior to cancer antigens due to their bio-stability and biological amplification [33,34] and can be detected months or years before clinical manifestation appears [35,36,37]. Indeed, somatic mutations and genomic instability result in large heterogeneity of the tumor cell proteome. Thus, the utilization of AAbs for cancer screening may present an opportunity to overcome biological heterogeneity, a major challenge in cancer biomarker research [11,38,39]. Although there are numerous studies about the identification of AAbs in the serum of cancer patients, including breast [40,41], colorectal [42,43,44], lung [45,46,47], ovarian [48,49,50], and gastrointestinal cancer [51,52,53,54], there is no study to address placental proteome as a target for monitoring cancer-associated AAbs. Analogies between placentation, in particular the behavior of trophoblast cells, and cancer raised our hypothesis on whether cancer patients develop autoantibodies against the placenta, which can be exploited as a tool for non-invasive cancer screening [55,56,57].

We observed that patients with BrC, CRC, and GC produce placenta-reactive autoantibodies in their serum when compared with age- and sex-matched controls. The placental reactivity of cancer sera supports the concept of a shared antigenic signature of cancer and placental cells and is in favor of the positive pleiotropy hypothesis. It should be noted that breast epithelial cells originate from the fetal ectoderm, while those of the gastrointestinal tract and bladder all originate from the fetal endoderm. Our findings, therefore, imply that although the placenta is not derived from embryonic germ layers, it has common antigens with them. Of note, sera from BC patients did not exhibit reactivity with placental antigens. The exact reason for this finding is not clear to us at present. However, the polarization of immune responses within the bladder tumor by a heavily immunosuppressive tumor microenvironment is a hallmark of this type of cancer [58], which could negatively affect the production of cancer-associated antibodies.

Interestingly, the reactivity of BrC sera with FTP was more pronounced compared with that of TP. Trophoblasts of the first-trimester placenta show a more invasive pattern compared to those of TP. The reason for this preferential reactivity of cancer sera with FTP may stem from the proteins that are differentially expressed in FTP and TP. FTP-expressed proteins were found to be mostly involved in response to stress, programmed cell death, cellular oxidant detoxification, and coagulation, the mechanisms that are also active in cancer development and the propagation of homeostasis [28]. To explore which cells in the placenta are the potential targets of cancer-associated autoantibodies, a human-on-human IHC was set using different sources of sera as primary antibodies. An intriguing finding was the clear reactivity of BrC sera with a small population of cells mostly residing in the trophoblast layer of the placental villi. Although reactivity was strong, the reactive cells were scarce, accounting for less than 1 in 1000 nucleate cells. The same population, but at a considerably lower frequency, was also observed when placentas were stained with sera from non-BrC women, which is in line with what we observed in ELISA. The exact nature of these cells is unclear at present, but they may represent a subpopulation of trophoblast stem cells (TS). Human TS have been generated from cytotrophoblasts and they have been shown to have the capacity to give rise to the three major trophoblast lineages. Interestingly, human TS cells injected into mice mimic trophoblast invasion during implantation [59]. First-trimester placenta cytotrophoblast cells are able to generate new villi, suggesting that villi cytotrophoblast cells may be a stem cell source [60]. Human TS cells cannot be derived from cytotrophoblast cells isolated from term placentas [57]. TS is a source for the generation of invasive cytotrophoblasts and this may explain the cross-reactivity of cancer-associated autoantibodies with these cells. Another finding was a significantly higher frequency of reactive cells with early-stage versus late-stage breast cancer sera. This finding may be attributed to immune cell exhaustion in the course of advanced breast cancer [61]. When tested by ELISA, sera from early-stage BrC and CRC showed higher reactivity with FTP than those from late-stage cancer sera, while the opposite was the case for GC patients, implying differential kinetics of anti-cancer humoral immune responses in different types of cancer.

Regarding the fact that some women in the patient and control groups had a history of previous pregnancy and placental exposure, we next examined the potential effect of the previous history of pregnancy or ongoing pregnancy on serum reactivity with placental antigens. There are conflicting data on the link between pregnancy and breast cancer risk [62,63,64,65,66]. It is reported that an early pregnancy before the age of 20 years could reduce the chance of developing breast cancer by 50% [67,68]. Further pregnancies reduce the likelihood of breast cancer by 10% [69]. In one cohort study conducted on 2.3 million Danish women, it was observed that full-term pregnancies lasting 34 weeks or longer reduce breast cancer risk, while the abortion of pregnancies lasting 33 weeks or less had no effects. The authors concluded that distinct biological effects around week 34 of pregnancy are a key point to understanding pregnancy-associated breast cancer protection [70]. It is noteworthy that women who have their first pregnancy after age 35 have a higher risk of developing breast cancer [67,68]. We observed that neither pregnancy history nor being pregnant at the time of blood sampling affect serum reactivity with either FTP or TP. One exception was the significantly higher reactivity of sera from multiparous pregnant women compared to nulligravida women with TP. Importantly, no difference was observed between multiparous non-pregnant and nulligravida women, indicating that it is the memory immunological responses that increase placenta-reactive autoantibodies in multiparous pregnant women. Indeed, the absence of a significant reactivity of sera from multiparous women with FTP may stem from a state of a tolerogenic microenvironment, which exists in uterine lymph nodes during the first-trimester of pregnancy [71,72]. We observed a decreasing trend of the reactivity of sera from first-trimester to full-term pregnancy sera with FTP, while this trend was increasing when TP was used as the antigen. This finding may imply the generation of placenta-specific autoantibodies during pregnancy, although comparisons did not show a significant difference. Indeed, it is interesting to note that autoantibodies in multiparous women were mostly directed against antigens of TP, while in cancer patients, serum reactivity was mostly directed against the proteome of FTP. This finding clearly shows that the immune presentation and recognition of placental antigens during pregnancy and cancer are mechanistically different. In line with this notion, we observed that although in a mixed population of BrC patients and non-BrC controls, women with ≥1 pregnancy had higher placenta-reactive AAbs compared to nulligravida women, the elevated levels of placenta-reactive AAbs in BrC patients is independent of a previous history of pregnancy and abortion. It is possible that the lower risk of BrC in women with a history of pregnancy is due to the cross-reactive T cells primed during pregnancy.

Limitations of the current study include: (1) we do not know exactly which protein or proteins are responsible for the generation of placenta-reactive AAbs in cancer patients, which is the prospect of our next study. (2) The sample size of GC, BC, and CRC was relatively small. (3) Although we showed the reactivity of breast cancer sera with scattered cells with cytotrophoblast morphology, the identity of these cells is unclear to us. 

## 5. Conclusions

In conclusion, our results showed that patients with cancer produce detectable levels of placenta-reactive AAbs. This is extremely the case for BrC, in which patients with early-stage disease produced AAbs reactive with the first-trimester placenta and specifically with a scattered subpopulation of cells, probably trophoblast stem cells. To our knowledge, the concept of the early screening of cancers using placental proteins has not been introduced before. This finding could shed light on the future application of placental proteome for the non-invasive early detection of cancer. This, in turn, will open new strategies to find candidate molecules for the targeted immunotherapy of cancer patients.

## Figures and Tables

**Figure 1 biomedicines-11-00316-f001:**
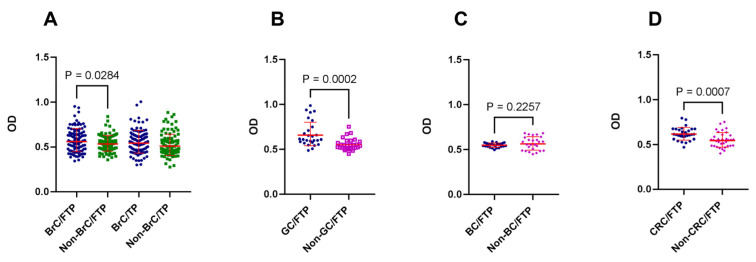
Immune reactivity of cancer-associated autoantibodies with first-trimester and term placental proteins: Reactivity of sera from breast (BrC) (**A**), gastric (GC) (**B**), bladder (BC) (**C**), and colorectal (CRC) (**D**) cancer patients against placental proteins were tested using ELISA. Sera from corresponding non-cancer age- and sex-matched controls were tested in parallel. In the case of BrC patients, the reactivity of sera with first-trimester (FTP) and term placental proteins (TP) was compared.

**Figure 2 biomedicines-11-00316-f002:**
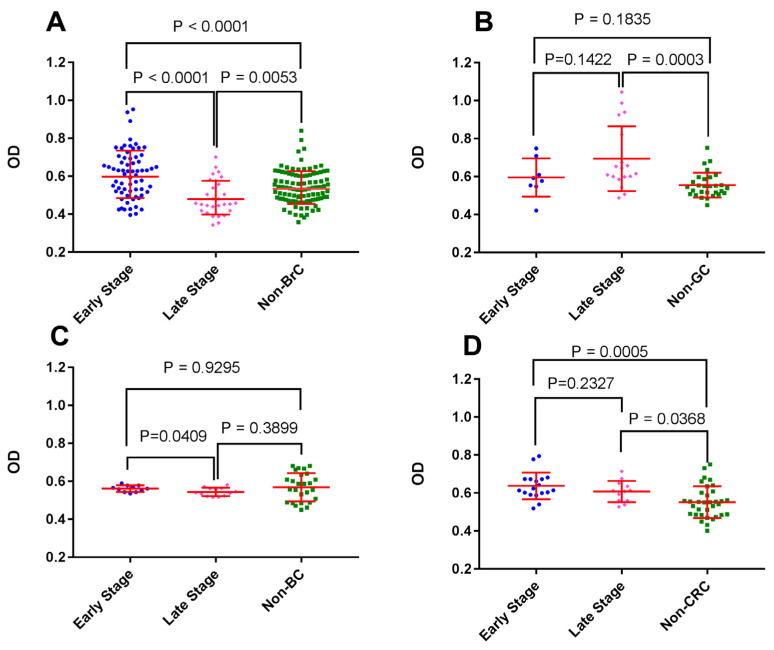
Analysis of the effect of cancer stage on serum reactivity with FTP proteins: Reactivity of sera from breast (BrC) (**A**), gastric (GC) (**B**), bladder (BC) (**C**), and colorectal (CRC) (**D**) cancer patients at early and late stages against FTP was tested using ELISA. Sera from corresponding non-cancer age- and sex-matched controls were tested in parallel.

**Figure 3 biomedicines-11-00316-f003:**
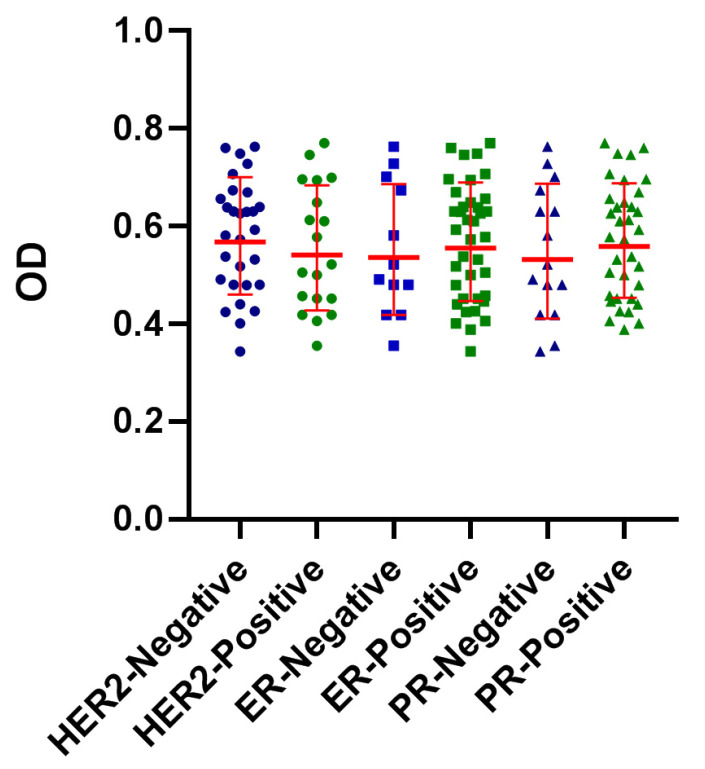
The effect HER2, ER, or PR expression on serum immune reactivity of BrC patients’ sera with FTP: Patients with breast cancer (BrC) were categorized based on the expression of estrogen receptor (ER), progesterone receptor (PR), and HER2 and the reactivity of their serum with first-trimester placental proteins (FTP) was tested.

**Figure 4 biomedicines-11-00316-f004:**
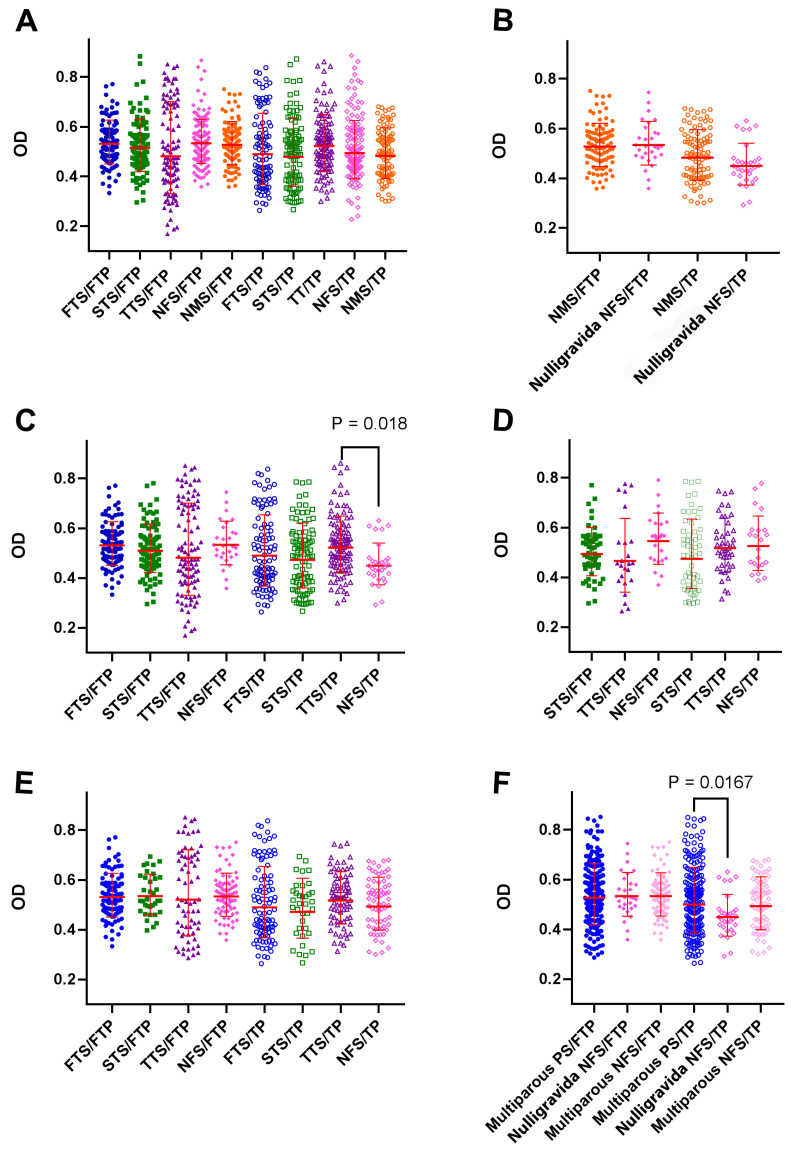
Effect of pregnancy and parity on serum reactivity with FTP and TP: (**A**) The reactivity of sera from pregnant women in the first (FTS), second (STS), and third trimester (TTS) of pregnancy with first-trimester (FTP) and term placental proteins (TP) was tested. Sera from non-cancer/non-pregnant (NFS) and normal males (NMS) served as controls. (**B**) The reactivity of nulligravida NFS against FTP and TP was compared with that of NMS. (**C**) The serum reactivity of nulligravida NFS and pregnant women at different trimesters of pregnancy with FTP and TP proteome was compared. (**D**) The reactivity of primiparous NFS and sera from primigravid pregnant women with FTP and TP proteomes was compared (collected samples contained no sera from primigravid women in first-trimester pregnancy). (**E**) The reactivity of multiparous NFS and sera from multiparous pregnant women at different trimesters of pregnancy with FTP and TP was compared. (**F**) The reactivity of nulligravida and multiparous NFS and sera from all multiparous pregnant women (regardless of pregnancy trimester) with FTP and TP was compared.

**Figure 5 biomedicines-11-00316-f005:**
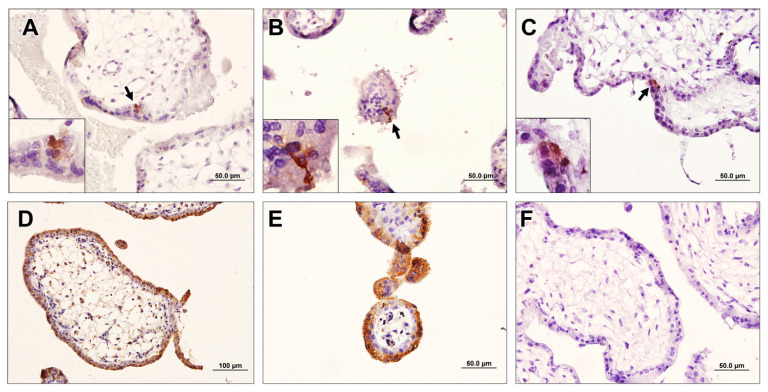
Immunohistochemical reactivity analysis of BrC sera with FTP: Sera from BrC (**A**,**B**) and non-BrC/non-pregnant (**C**) with first-trimester placenta (FTP) were tested with the human-on-human immunohistochemical staining. Hyperimmune serum from FTP-immunized rabbits was used as a positive control (**D**,**E**). Sera from normal males served as negative control (**F**). Black arrows show the positive cells.

**Table 1 biomedicines-11-00316-t001:** Case and control sample size and cancer stage distribution.

Cancer Type	Sample Size	Mean Age	Gender		Stage	
			Female	Male	Early	Late
CRC	34	51 ± 11.86	14	20	19	15
BrC	100	44 ± 8.41	100	0	70	30
BC	29	57 ± 14.08	10	19	13	16
GC	30	57 ± 10.44	14	16	10	20
Non-CRC	34	51 ± 11.84	14	20	-	-
Non-BrC	100	43 ± 8.97	100	0	-	-
Non-BC	29	56 ± 31.03	10	19	-	-
Non-GC	30	57 ± 10.28	14	16	-	-

**Table 2 biomedicines-11-00316-t002:** Demographic data of male and pregnant-female samples.

Cancer Type	Sample Size	Mean Age (Year)	Gestational Age (Weeks)
First-trimester pregnant women	100	33 ± 3.77	11–13
Second trimester pregnant women	100	32 ± 4.07	15–18
Third trimester pregnant women	110	29 ± 6.67	37–41
Normal males	100	38 ± 9.57	-

**Table 3 biomedicines-11-00316-t003:** Inclusion and exclusion criteria of cancer and non-cancer serum samples.

Groups	Inclusion Criteria	Exclusion Criteria
Case	Pathologically confirmed cancer	Current or past history of autoimmune diseases
	Available clinical pathological data	Previous history of cancer
	No history of any treatment (surgery, radiotherapy and/or chemotherapy, hormone therapy) before sampling	Received immunosuppressive drugs during the three months before sample collection
	Newly diagnosed cases	History of chemo, adjuvant, or radiotherapy and cancer surgery
Control	Age- and sex-matched subjects	Not age- and sex-matched
	Subjects with no history of cancer	History of chronic diseases or cancer
	Non-BrC females with confirmed non-malignant report on breast Fine Needle Aspiration (FNA)	Last pregnancy ≤ 3 years before cancer onset (female patients)
	Non-GC patients as confirmed by endoscopic gastric sampling and pathology confirmation	More than two abortions
	Non-CRC and non-BC as confirmed by colonoscopy and sonography	Current or past history of autoimmune diseases
	Normal males referred to medical laboratories for a routine checkup with normal biochemical and hematological test results	History of immunosuppressive drug intake three months before sampling
	Pregnant females in first or second trimesters of pregnancy with normal screening tests	Having no clinical pathological data
	Pregnant female in third trimester of pregnancy referred for cesarean section	More than 10 pregnancies

All control sera were age- and sex-matched compared to the cancer sera. The serum samples were kept at −20 °C in small aliquots until use and they were thawed only once before use.

**Table 4 biomedicines-11-00316-t004:** Linear regression analysis of 200 females in BrC and Non-BrC groups for gravidity and abortion.

Parameter	B	95% Wald Confidence Interval		Sig.
		Lower	Upper	
BrC	0.051	0.016	0.086	0.005
Non-BrC	Referent			
Gravidity ≥ 1	0.060	0.104	0.016	0.007
Gravidity = 0	Referent			
Abortion ≥ 1	0.001	−0.040	0.043	0.957
Abortion = 0	Referent			

## Data Availability

The datasets used and/or analyzed during the current study are available from the corresponding author upon reasonable request.

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
