# Peer review of "Cancer Is Associated with the Emergence of Placenta-Reactive Autoantibodies"

_biomedicines, 2023, doi:10.3390/biomedicines11020316_

Round 1

Reviewer 1 Report

1) Abstract. Placenta-specific antigens are minimally expressed or unexpressed in normal adult tissues, while they are widely expressed in cancer. In the course of carcinogenesis, a vast array of autoantibodies (AAbs) is produced. Here, we used a quantitative approach to determine the reactivity of AAbs in sera of patients with breast (BrC), gastric (GC), bladder (BC), and colorectal (CRC) cancers against first-trimester (FTP) and full-term placental proteome (TP) in comparison with the age- and sex-matched non-cancer individuals. Please, ameliorate the description of sample population.

2)  A human on human immunohistochemistry was set to determine reactive target cells in FTP. The effect of pregnancy on the emergence of placenta-reactive autoantibodies was tested using sera from pregnant women at different trimesters of pregnancy. Except for BC, patients with BrC (p<0.0284), GC (p<0.0002), and CRC (p<0.0007) had significantly higher levels of placenta-reactive AAbs. BrC (p<0.0001) and BC (p<0.0409) in the early stages triggered higher autoantibody reactivity against FTP. Reactivities of BrC sera with FTP did not show an association with ER, PR or HER2 expressions. Pregnancy in the third trimester was associated with the induction of TP- and not FTP-reactive autoantibodies (=0.0204). The reactivity of BrC sera with placental proteins was found to be independent of gravidity or abortion. BrC sera showed a very strong and specific pattern of reactivity with scattered cells beneath the syncytiotrophoblast layer. Our results showed that patients with cancer produced detectable levels of placenta-reactive AAbs, which could shed light on the future application of the placental proteome for the non-invasive early detection of cancer.. Please, ameliorate the conclusions and underline the clinical utility of the study.

3) Introduction. L51-54. Cancer cells and placenta share some char acteristics including immune escape and growth mechanisms as well as antigenic determinants. In order to discuss the previously described points, important references are needed to be added, such as:

a- SIRPγ-expressing cancer stem-like cells promote immune escape of lung cancer via Hippo signaling. J Clin Invest. 2022;132(5):e141797. doi:10.1172/JCI141797

b-The History and Mystery of Alveolar Epithelial Type II Cells: Focus on Their Physiologic and Pathologic Role in Lung. Int J Mol Sci. 2021;22(5):2566. Published 2021 Mar 4. doi:10.3390/ijms22052566.

4) L 92-96.To explore the potential usefulness of placenta-reactive AAbs for early diagnosis of  cancer, we performed a comprehensive analysis of these AAbs in sera of breast, gastric,  bladder, and colorectal cancer patients in comparison with age- and sex-matched non cancer controls. Identification of AAb-reactive target cells in the placenta and examining  the potential effect of pregnancy on the emergence of these AAbs were also among the  additional aims of this study. Please, improve the description of study aims.

5) Methods. L122-127. 1) Normal males referred to Farmand medical laboratory for a routine checkup (n =100,  mean age 38years ± 9.57 years), 2) Normal pregnant women at first and second trimesters  of pregnancy with normal screening tests referred to the department of Fetal and Maternal  Health in Hope Generation Foundation (HGF), Tehran, Iran (n =100, mean age 33 years ±  3.77 years for the first-trimester and n = 100, mean age 32 years ± 4.07years for the second trimester), 3) Normal female at the third trimester of pregnancy referred to the caesarian  section in Kamali hospital, Karaj, Iran (n = 110, mean age 29 years ± 6.67 years). Please clarify the sample population

6) 4. Discussion L326-331. The theory of the fetal origin of cancer has long been proposed and many supporting evidence has been reported [27]. A very fascinating connection between placentation and  carcinogenesis has been suggested by Wagner et. al. Accordingly, a link between species specific malignancy rate and placentation type is likely related to the level of adapted Cis Regulatory mechanisms in the endometrium that controls endometrial resistance to invis ibility in eutherian mammals [28]. Please, summarise here the most important results of the study.

7) 4. Discussion. Please, underline the limits of the study.

8) 5. Conclusions. Please underline the novelty of the study and the clinical implications. 

Author Response

I would like to thank the reviewer for the constructive comments. We did our best to revise the paper according to the comments raised by the reviewer. Below please find point by point response of the authors to the reviewer’s comments. I hope this revision take the positive opinion of the respected editor and reviewer for acceptance of our manuscript.

The best regards

Amir-Hassan Zarnani 

Placenta-specific antigens are minimally expressed or unexpressed in normal adult tissues, while they are widely expressed in cancer. In the course of carcinogenesis, a vast array of autoantibodies (AAbs) is produced. Here, we used a quantitative approach to determine the reactivity of AAbs in sera of patients with breast (BrC), gastric (GC), bladder (BC), and colorectal (CRC) cancers against first-trimester (FTP) and full-term placental proteome (TP) in comparison with the age- and sex-matched non-cancer individuals. Please, ameliorate the description of sample population.

  • Description of sample population is now mentioned in abstract.

A human on human immunohistochemistry was set to determine reactive target cells in FTP. The effect of pregnancy on the emergence of placenta-reactive autoantibodies was tested using sera from pregnant women at different trimesters of pregnancy. Except for BC, patients with BrC (p<0.0284), GC (p<0.0002), and CRC (p<0.0007) had significantly higher levels of placenta-reactive AAbs. BrC (p<0.0001) and BC (p<0.0409) in the early stages triggered higher autoantibody reactivity against FTP. Reactivities of BrC sera with FTP did not show an association with ER, PR or HER2 expressions. Pregnancy in the third trimester was associated with the induction of TP- and not FTP-reactive autoantibodies (=0.0204). The reactivity of BrC sera with placental proteins was found to be independent of gravidity or abortion. BrC sera showed a very strong and specific pattern of reactivity with scattered cells beneath the syncytiotrophoblast layer. Our results showed that patients with cancer produced detectable levels of placenta-reactive AAbs, which could shed light on the future application of the placental proteome for the non-invasive early detection of cancer. Please, ameliorate the conclusions and underline the clinical utility of the study.

  • Description of the conclusions and the clinical utility of the study are now mentioned in the abstract. Due to the limitation of word count in the abstract, It was only mentioned in one sentence

L51-54. Cancer cells and placenta share some characteristics including immune escape and growth mechanisms as well as antigenic determinants. In order to discuss the previously described points, important references are needed to be added, such as:

a- SIRPγ-expressing cancer stem-like cells promote immune escape of lung cancer via Hippo signaling. J Clin Invest. 2022;132(5):e141797. doi:10.1172/JCI141797

b-The History and Mystery of Alveolar Epithelial Type II Cells: Focus on Their Physiologic and Pathologic Role in Lung. Int J Mol Sci. 2021;22(5):2566. Published 2021 Mar 4. doi:10.3390/ijms22052566.

  • Suggested references are now added in the introduction.

L 92-96.To explore the potential usefulness of placenta-reactive AAbs for early diagnosis of cancer, we performed a comprehensive analysis of these AAbs in sera of breast, gastric,  bladder, and colorectal cancer patients in comparison with age- and sex-matched non cancer controls. Identification of AAb-reactive target cells in the placenta and examining  the potential effect of pregnancy on the emergence of these AAbs were also among the  additional aims of this study. Please, improve the description of study aims.

  • Description of the aim of the study was revised and improved.

L122-127. 1) Normal males referred to Farmand medical laboratory for a routine checkup (n =100,  mean age 38years ± 9.57 years), 2) Normal pregnant women at first and second trimesters  of pregnancy with normal screening tests referred to the department of Fetal and Maternal  Health in Hope Generation Foundation (HGF), Tehran, Iran (n =100, mean age 33 years ±  3.77 years for the first-trimester and n = 100, mean age 32 years ± 4.07years for the second trimester), 3) Normal female at the third trimester of pregnancy referred to the caesarian  section in Kamali hospital, Karaj, Iran (n = 110, mean age 29 years ± 6.67 years). Please clarify the sample population

  • A table describing demographic data of male and pregnant female samples was provided.

Discussion L326-331. The theory of the fetal origin of cancer has long been proposed and many supporting evidence has been reported [27]. A very fascinating connection between placentation and carcinogenesis has been suggested by Wagner et. al. Accordingly, a link between species specific malignancy rate and placentation type is likely related to the level of adapted Cis Regulatory mechanisms in the endometrium that controls endometrial resistance to invis ibility in eutherian mammals . Please, summarise here the most important results of the study.

  • According to the reviewer’s suggestion, a section describing the most important results of the study was added.

Please, underline the limits of the study.

  • Limitation of study was added and at the end of the discussion part.

Please underline the novelty of the study and the clinical implications. 

  • As per reviewer’s instruction, novelty and potential clinical application of this study was underlined at the end of manuscript.

Reviewer 2 Report

The manuscript submitted by Sarvestani et al. entitled "Cancer is associated with the emergence of placenta-reactive autoantibodies" to the Special Issue "Selected Papers in the 2nd International Electronic Conference on Biomedicines (ECB 2023)" aims to uncover if cancer-patients produce antibodies that can recognize placental cells/proteins.

The manuscript is well-written generally speaking. However, regarding the distribution of the pictures among the manuscript, the figures should be placed near the text when they are cited at the first time.

Additionally, in the M&M section, the authors should fully describe the used methodologies instead of making citation of additional references.

The results are well discussed. However, the authors should make some additional effords to characterize the cellular subpopulation in the placentas that are recognized by the antibodies produced by the cancer patients (RNAseq, IHC??). Authors also should clarify why they used the central zone of the placentas and not other regions.

The p-values in the figures 3 and 4 should be indicated.

Author Response

I would like to thank the reviewer for the constructive comments. We did our best to revise the paper according to the comments raised by the reviewer. Below please find point by point response of the authors to the reviewer’s comments. I hope this revision take the positive opinion of the respected editor and reviewer for acceptance of our manuscript.

The best regards

Amir-Hassan Zarnani 

The manuscript is well-written generally speaking. However, regarding the distribution of the pictures among the manuscript, the figures should be placed near the text when they are cited at the first time.

  • We agree with reviewer’s comment. However, the figures have been applied in part 3.7 based on the standard format of Biomedicine journal. The journal will finally place the figures in the right positions.

Additionally, in the M&M section, the authors should fully describe the used methodologies instead of making citation of additional references.

  • Revision was made according to the reviewer’s suggestion. More explanations are now included in the text, which can give a comprehensive view of the process.

The results are well discussed. However, the authors should make some additional efforts to characterize the cellular subpopulation in the placentas that are recognized by the antibodies produced by the cancer patients (RNAseq, IHC??).

  • The authors are thankful for this meticulous point of the reviewer’s view. In fact, this was also our concern to specifically uncover the identity of the reactive placental cells. However, characterization of these cells using a panel of antibodies or isolation of the reactive cells to perform single cell RNAseq is costly, which is the main limitation of our project. As a shortcut for this limitation, although not precise, reactive cells was carefully examined by an expert pathologist and confirmed to be from cytotrophoblastic lineage. This limitation is now added to the end of discussion section.

Authors also should clarify why they used the central zone of the placentas and no other regions.

  • This question may arise from our unclear description of the methodology. In fact, we collected five punches (each one gram) from different parts of each placenta, four punches from four locations (including both maternal and fetal sides) and one punch from the central part of the placenta. In this way, we tried to minimize the potential spatial variation that may exist in placental proteome. This explanation is now clear in M&M section.  

The p-values in the figures 3 and 4 should be indicated.

  • There were many comparisons that could be made. Regarding the fact that a majority of comparisons were not statistically significant and to avoid complexity of the figures, only significant p-values were indicated. In case, when the respected reviewer insists on this revision, the authors are ready to provide all p values in a supplementary table.

Reviewer 3 Report

In their paper, the authors presented how cancers are associated with the production of placenta-reactive antibodies and how this could help in the early detection of cancers.

I really have no complaints. The entire paper is very clear and well presented. All parts are well written.

Author Response

I would like to thank the reviewer comment. Below please find response of the authors to the reviewer’s comments. I hope this revision take the positive opinion of the respected editor and reviewer for acceptance of our manuscript.

The best regards

Amir-Hassan Zarnani 

In their paper, the authors presented how cancers are associated with the production of placenta-reactive antibodies and how this could help in the early detection of cancers.

I really have no complaints. The entire paper is very clear and well presented. All parts are well written.

  • The authors would like to thank this reviewer for the heartwarming scoring and comments.

Round 2

Reviewer 2 Report

Thank you for answered to all raised questions.